# Pharmacy Workplace Wellbeing and Resilience: Themes Identified from a Hermeneutic Phenomenological Analysis with Future Recommendations

**DOI:** 10.3390/pharmacy10060158

**Published:** 2022-11-23

**Authors:** Jon C. Schommer, Caroline A. Gaither, Nancy A. Alvarez, SuHak Lee, April M. Shaughnessy, Vibhuti Arya, Lourdes G. Planas, Olajide Fadare, Matthew J. Witry

**Affiliations:** 1Department of Pharmaceutical Care & Health Systems (PCHS), College of Pharmacy, University of Minnesota, 308 Harvard Street SE, Minneapolis, MN 55455, USA; 2R. Ken Coit College of Pharmacy–Phoenix, University of Arizona, 650 East Van Buren Street, Phoenix, AZ 85004, USA; 3American Pharmacists Association, 2215 Constitution Avenue NW, Washington, DC 20037, USA; 4College of Pharmacy and Health Sciences, St. John’s University, 8000 Utopia Parkway, Queens, New York, NY 11439, USA; 5College of Pharmacy, University of Oklahoma, 1110 N Stonewall, Room 243, Oklahoma City, OK 73117, USA; 6College of Pharmacy, University of Iowa, 180 South Grand Avenue, Iowa City, IA 52242, USA

**Keywords:** pharmacist, technician, pharmacy, stress, burnout, distress, moral injury, workism, trauma, responsibility, autonomy, subjection, sociocultural, systems, hermeneutic phenomenology

## Abstract

This study applied a hermeneutic phenomenological approach to better understand pharmacy workplace wellbeing and resilience using respondents’ written comments along with a blend of the researchers’ understanding of the phenomenon and the published literature. Our goal was to apply this understanding to recommendations for the pharmacy workforce and corresponding future research. Data were obtained from the 2021 APhA/NASPA National State-Based Pharmacy Workplace Survey, launched in the United States in April 2021. Promotion of the online survey to pharmacy personnel was accomplished through social media, email, and online periodicals. Responses continued to be received through the end of 2021. A data file containing 6973 responses was downloaded on 7 January 2022 for analysis. Usable responses were from those who wrote an in-depth comment detailing stories and experiences related to pharmacy workplace and resilience. There were 614 respondents who wrote such comments. The findings revealed that business models driven by mechanized assembly line processes, business metrics that supersede patient outcomes, and reduction of pharmacy personnel’s professional judgement have contributed to the decline in the experience of providing patient care in today’s health systems. The portrait of respondents’ lived experiences regarding pharmacy workplace wellbeing and resilience was beyond the individual level and revealed the need for systems change. We propose several areas for expanded inquiry in this domain: (1) shared trauma, (2) professional responsibility and autonomy, (3) learned subjection, (4) moral injury and moral distress, (5) sociocultural effects, and (6) health systems change.

## 1. Introduction

### 1.1. Pharmacy Workplace Wellbeing and Resilience

Pharmacy personnel’s workplace issues and their relationship to personal wellbeing continue to be critical, complex issues across all practice settings [1], and have been further exacerbated and exposed under the COVID-19 pandemic [2,3,4,5,6,7]. In addition to burnout, stressful pharmacy job demands have been linked to patient safety concerns, especially medication errors [8,9].

Even before the COVID-19 pandemic, pharmacy personnel in the United States were experiencing high levels of stress due to the U.S. health care system’s shift from a public service to a business model that began in the latter half of the 20th century [10]. Pharmacy practice involves intimate caregiving relationships that often require the suspension of routine pharmacy operations in order to address serious patient-specific needs [9,10]. Business models driven by mechanized assembly line processes, business metrics that supersede patient outcomes, and reduction of pharmacy personnel’s professional judgement have contributed to the decline in the experience of providing patient care in today’s health systems [10,11,12].

Traditional approaches for addressing stress and burnout are to (1) identify them using sound measures and (2) build personal resilience as a key requisite for coping. Regarding identification and measurement, useful models of job stress and burnout have been developed and applied for identifying factors that cause stress, the outcomes of such stress, and ways to manage and prevent stress [3,4,5,6,7,13,14,15,16]. A recent integrative review of burnout and stress literature in pharmacy identified 491 articles that used 11 psychometrically sound measures [15]. Regarding personal resilience, it is the most often identified way to cope with stress [12,15,17,18]. In addition to a focus on individual responsibility for resilience, more expansive stress management and prevention literature is beginning to emerge in pharmacy [19]. For example, recent research in pharmacy [9,11] suggests that the COVID-19 pandemic was a shared crisis that extended beyond individual and organization levels, to the health care system itself, and in some cases, at a sociocultural level [9,11,12]. Thus, ways to cope with stress may have broadened as well from the individual and organizational level to health systems and societal levels. Some examples of these are presented next.

### 1.2. Health Systems and Moral Distress or Moral Injury

The U.S. health care system’s shift from a public service to a business model [10] pressured pharmacy to meet efficiency challenges through market power dynamics, negotiated contracts, pay-for-performance incentives, and corporate-level metrics. However, top-down, rigid, business-centric management decisions created dissonance for pharmacy personnel’s ability to provide patient-centered care, exercise professional judgement, and experience joy and meaning in providing care [9,11,12]. Two phenomena, moral distress and moral injury, may be useful to understand the dissonance expressed by some in their daily work. Moral distress is a “psychological disequilibrium which occurs when a provider is able to make a moral judgment about the correct choice, but is not able to provide the care that is perceived to be ‘right’ or ‘best’ for the patient” [20]. As Talbot and Dean [21] wrote about developing moral injury:


*“Navigating an ethical path among such intensely competing drivers is emotionally and morally exhausting … Routinely experiencing the suffering, anguish, and loss of being unable to deliver the care that patients need is deeply painful. These routine, incessant betrayals of patient care and trust are examples of ‘death by a thousand cuts.’ Any one of them, delivered alone, might heal. But repeated on a daily basis, they coalesce into the moral injury of health care.”*


These health system effects already were detected before the COVID-19 pandemic [1,12], but the onset of the pandemic exposed and amplified the likely experiences of moral distress and moral injury for pharmacy personnel.

### 1.3. The Sociocultural Phenomenon of Workism

In addition, a sociocultural phenomenon called ‘workism’ has affected pharmacy personnel wellbeing and resilience [12,22,23] by building an expectation that work is the centerpiece of one’s identity and life’s purpose. Thompson states that “a culture that funnels its dreams of self-actualization into salaried jobs is setting itself up for a collective anxiety, mass disappointment, and inevitable burnout” [22]. The majority of pharmacists in the United States are now under the age of 40 with less than 15 years of experience after licensure [1]. This group reports higher stress and burnout than those with more experience [1,5,14]. They passed through a childhood of extracurricular achievement, were told that their work should be their passion, took on high levels of student debt, and face the disturbance of social media which amplifies the pressure to project a superficial image of success to others [22]. Further, social media use is linked to dehumanization, anxiety, stress, and relational problems [23]. These sociocultural effects already were detected before the COVID-19 pandemic [1,12], but the onset of the pandemic exposed and amplified the notion of workism for pharmacy personnel.

### 1.4. Application of a Hermeneutic Phenomenological Approach

As we completed research in the pharmacy workplace and resilience domain, aspects of moral injury and workism emerged in the findings [9,11,12]. This led us to consider that relying on personal resilience as a key requisite for coping with stress and burnout is not sufficient and may actually harm individuals from thinking they are at sole fault for experiencing burnout. Health system and sociocultural effects were reported as contributors to stress, burnout, and feelings of despair since resilience could not be achieved within the situations being faced. In the 2021 American Pharmacists Association (APhA) and National Alliance of State Pharmacy Associations (NASPA) National State-Based Pharmacy Workplace Survey, a series of five open-ended questions was used to learn about respondents’ opinions and experiences in their own words. The first four questions focused on barriers and facilitators to their ability to perform duties for optimal patient care and/or ensuring patient safety. Findings from these questions have been published [9,11].

The fifth question simply asked if respondents had any other comments. We were surprised to find that 1327 (19%) out of 6973 responders took the time to write additional comments. Following guidance from phenomenology research [24,25,26,27,28,29], we included comments for analysis if the comments included information about the respondent’s “lived experience” [25,26]. Applying this approach resulted in 614 (46%) comments that were in-depth responses that described stories and experiences. The comments that did not meet our inclusion criteria (the other 54%) mostly related to thanking us for conducting the survey with some comments also giving suggestions about how to change pharmacy or healthcare. In order to gain a better understanding about workplace wellbeing and resilience, we turned to a hermeneutic phenomenological approach as a way to appreciate respondents’ lived experiences within their workplaces as described in their own words. This approach seeks to uncover meaning within the experience by analyzing narratives about the experience [24,25,26,27,28,29]. The 614 in-depth stories were well suited for such an approach.

### 1.5. Study Objectives

This study applied a hermeneutic phenomenological approach to better understand pharmacy workplace wellbeing and resilience. Rather than a focus on the causes of the phenomenon, our aim was to present a portrait of the respondents’ lived experiences. The objective of our inquiry was to describe pharmacy workplace wellbeing and resilience using respondents’ written comments along with a blend of the researchers’ understanding of the phenomenon and the published literature [28,29]. Our goal was to apply this understanding to recommendations for the pharmacy workforce and corresponding future research.

## 2. Materials and Methods

### 2.1. Data Source

Data were obtained from the 2021 APhA/NASPA National State-Based Pharmacy Workplace Survey [9], launched nationally in April 2021 by the two organizations. Promotion of the online survey to pharmacy personnel was accomplished through social media, email, and online periodicals. Responses continued to be received through the end of 2021. A data file containing 6973 responses was downloaded on 7 January 2022 for analysis.

For the purpose of this study, usable responses were from those who wrote an in-depth comment detailing stories and experiences related to pharmacy workplace wellbeing and resilience. There were 614 respondents who wrote such comments, and these were used for analysis.

### 2.2. Hermeneutic Phenomenological Analysis

The 614 in-depth comments were stored in a Word file and distributed to five research team members (NA, CG, SL, AS, JS). Written comments were read several times by each person independently. The main stories, anecdotes, and insights written by the respondents were analyzed through van Manen’s “inceptual process” of reflective wondering, deep questioning, attentive reminiscing, reflective epoché, and sensitively interpreting primal meanings of experiences [25,26]. We did this without imposing judgement and with self-reflexivity and recognition about our own experiences and beliefs. We focused on the respondents’ lived experiences and their ways of describing these experiences.

An example of this method is from a study of patients with depression [29]. Focusing on the causes of depression was deemed insufficient for understanding a portrait of the experience of depression itself. In that study, people described depression as a place that takes the person away from their “everydayness (alltaglichkeit) and their homelikeness (heimlichkeit)” to a place of “uncanniness and un-homelikeness”. Depression was described using figurative spatial imagery such as constriction, bottomless pit of darkness, wilderness or isolated places. We applied a similar approach to understanding a portrait of workplace wellbeing and resilience.

In our first step of analysis, we asked “What do the comments tell us about respondents’ lived experiences and how can they be described?” After this initial reading, the five team members met in March 2022 and identified issues like (1) desperation, (2) imagery of dying, (3) drowning, (4) moral injury, (5) hopelessness, and (6) regret. The initial consensus was that themes were consistent with the workplace burnout literature. After this meeting, each team member independently read the comments again and we reconvened in April 2022. During that same time, one team member (SL) conducted a literature review of the stress and burnout literature.

During the next team meeting, the same themes emerged, but discussion about the inability of pharmacy personnel to meet all of their expected responsibilities for their organization, profession and patients emerged. Comments were identified that went beyond the stress and burnout literature that focused on individual characteristics and rather, were about the overall health care system and sociocultural influences. This prompted the team to search the moral injury and workism literature, and also to meet with an expert in the area of pharmacist responsibilities (LP) for further discussion. This helped broaden the scope of descriptions into areas of responsibility, professionalism, health system effects, and sociocultural effects. During this meeting, the group expanded discussions into areas such as moral distress, learned helplessness, ‘metrics equal worth’, practice isolation, coaching/mentoring, perceived responsibility, just cultures, joy and meaning in work, and improving the experience of providing care.

In April 2022, one team member (NA) took the list of identified elements and began grouping lived experience exemplars for each element. As this was being completed, the other four team members consulted the literature and shared ideas with the whole team. In late April 2022, the team met again and reviewed the elements and exemplars that were identified. In May 2022, the team met to discuss elements and exemplars in order to reveal inceptual insights [25,26]. On June 16, 2022 the team members agreed upon 15 elements, with relevant exemplars and insights. At this point (June 2022), the team grouped the 15 elements into six (6) categories for review and discussion. These are described in the Appendix A (see Table A1).

In addition to the five research team members (NA, CG, SL, AS, JS), four additional research experts (VA, OF, LP, MW) were invited to review the elements and categories and develop recommendations for expanding pharmacy workforce wellbeing and resilience research. Each co-author focused on one of six categories:Shared Trauma (VA)Threats to Professional Responsibility and Autonomy (LP)Learned Subjection (CG)Moral Distress and Moral Injury (NA, AS)Sociocultural Effects (SL, JS)Plea for Health Systems Change (OF, MW)

### 2.3. Research Team and Reflexivity

As hermeneutic phenomenological analyses were completed, team member reflexivity was conducted so that assumptions were acknowledged and documented as part of the research process [30,31,32]. The research team for this paper consisted of nine people (NA, VA, OF, CG, SL, LP, AS, JS, MW). All nine have experience in pharmacist workforce and quality of work life research. Four members (NA, VA, SL, MW) hold PharmD degrees and have experience in advanced clinical care practice. Four members (CG, LP, JS, MW) hold PhD degrees in pharmacy. Four members (NA, VA, LP, AS) are actively engaged in legislative policy and advocacy work. All nine team members hold licenses to practice pharmacy. Each member of the team interacts with pharmacists and student pharmacists on a regular basis. Each member of the team has work experience in pharmacy practice and these experiences were shared during analytic discussions. During analysis, team members acknowledged how they were being affected by personal events that included learning about and supporting others through injustice, inequity, mental health, and dealing with suicide. Our team acknowledged how these events could impact our analysis and interpretation of the findings. The background of the research team provides strengths to this project that helped analyze and interpret the data collected. Personal presuppositions from both professional and personal experiences were noted and accounted for in how the analysis may have been influenced.

### 2.4. Rigor

Realism was supported by reaching congruence among research team members regarding the data‘s relevance, sense, and accuracy and the use of thick descriptions [31]. Credibility and authenticity were reinforced by multiple readings of the text, member checking, and documentation [27,31,32,33,34]. This helped assure that data interpretation echoed the respondents’ words and not the biases and viewpoints of the research team [27,31,32]. Audit trails were maintained in order to document each aspect of the research process [31,32].

## 3. Results

Fifteen elements emerged from the data that described pharmacy workplace wellbeing and resilience using respondents’ written comments along with a blend of the researchers’ understanding of the phenomenon and the published literature [28,29]. These 15 elements were grouped into six categories as follows:Shared Trauma (Helplessness; De-professionalization of pharmacy)Threats to Professional Responsibility and Autonomy (Disempowerment, Power-dependence)Learned Subjection (Oppression; Abandonment; Depersonalization; Dehumanization)Moral Distress and Moral Injury (Moral distress; Moral injury)Sociocultural Effects (Despair; Disdain)Plea for Health Systems Change (Different cultures; Now is the time; Plea for change)

Written exemplars for each element are presented in the Appendix A. It is noteworthy that the 15 elements and six categories are negative in nature. There were very few positive comments and these typically were about being thankful where they worked so that they could avoid one of the 15 elements we identified. The results show that the portrait of respondents’ lived experiences regarding pharmacy workplace wellbeing and resilience are beyond the individual level. There is shared malaise surrounding a collective experience of helplessness, loss of professional standing, oppression, moral injury, workism, and a plea for rescue. The results show the need for health care system and societal change and provide insights for future research. This will be presented in the discussion section of this paper.

## 4. Discussion

### 4.1. Limitations

Before the findings are discussed, several limitations should be considered. The results did not use a random sample of pharmacy personnel. Thus, the findings should be used for gaining insight and not be used for making estimates for or to generalize to the entire population of pharmacy personnel. Not all survey respondents provided written comments. It is likely that those who wrote comments had strong opinions or were interested in the topic. The majority of comments came from pharmacy personnel working in community practice. The distribution of respondents to the survey and associations between respondent types and patterns of response may be found in the full report [9].

### 4.2. Structure Used for the Discussion

The findings described pharmacy workplace wellbeing and resilience using survey respondents’ written comments along with a blend of the researchers’ understanding of the phenomenon and the published literature [28,29]. This portrait of the respondents’ lived experiences was used as a guide for us to develop recommendations for future research in this domain. The discussion is presented in six sections that describe proposed areas for future research.

### 4.3. Recommendations for Expansion of Research in Pharmacy Workplace Wellbeing and Resilience

#### 4.3.1. Shared Trauma

The experience of COVID-19, much like any other natural disaster or major event that communities experience collectively (e.g., hurricanes, 9/11/01 attacks in New York City, etc.) upended lives of millions and disrupted nearly every individual’s life across the globe. The extent of this disruption varied across communities, ranging from minor inconveniences to loss of lives, economic hardship, and long-term negative health consequences. Existing inequities were magnified, cracks across systems revealed in more pronounced manners, and impact on well-being and behavioral health emerged in a much stronger way across society. No matter where on the spectrum of disruption the individual experience of this trauma fell, communities of people were all collectively feeling the impact of this extraordinary situation. Particularly for healthcare professionals, this time has marked a shared trauma that both patients and their care providers were experiencing at the same time.

Shared trauma is “the affective, behavioral, cognitive, spiritual, and multi-modal responses that clinicians experience as a result of dual exposure to the same collective trauma as their clients” [35]. It is important to note that while persons providing care are impacted by the same trauma as patients, their response may not necessarily be the same. As a result of this shared trauma, the care providers’ thoughts, perceptions, worldview, response to stress, coping, and ability to provide care while experiencing these unique stresses collectively can be impacted. In particular, without time to process and develop coping mechanisms to promote healing and resilience, care providers can experience burnout quicker and easier [35]. Comments to the survey revealed how pharmacy personnel suffered from a shared malaise that came with feelings of being punished, taken advantage of, and used up to the point of utter defeat:


*“The main issue is the whole structure of pharmacy as a retail business with a small number of large corporations controlling the services we provide, how they are provided and how much is made in those service with the profits going to stock holders not the healthcare providers who are providing the services. The expectation is to treat numbers not patients. We are punished when a patient does not take a medication they do not want to take. We are punished when a fellow healthcare professional does not feel comfortable prescribing a medication for a patient for legitimate reasons. We are punished if a patient gets hospitalized and does not take their medication for two weeks. EQUIPP scores are used by PBMs to maximize punishment not reward.”*



*“Chain pharmacy has become nothing more than factory work. Instead of making parts, it’s filling prescriptions as fast as you can all day long. It’s soul killing. Caring for patients? It’s more like caring for their pocketbooks. PBMs add another layer of despicable behavior to their mix.”*


In order to heal from shared trauma, mechanisms on a systems level must be developed and implemented so that individuals can create the time and space necessary to process their relationship to shared trauma and have or maintain the capacity to care for patients in a compassionate manner [35]. Systems interventions that allow for support, time, and training are necessary to help mitigate the harm from shared trauma. While it is important for individuals to work on individual resilience, stress responses and self-care, it is also crucial to critically examine systems and how we treat people in the system.

It is clear that pharmacy personnel collectively know about the shared trauma that they and their patients are experiencing. However, they are suffering and grieving in isolation without any hope of finding a way out. We propose that there is a need for action to identify and prioritize the needs of people within systems rather than seeing and treating them as a means to meet productivity metrics. It is essential to understand their shared trauma and offer functional and healthy systems-level solutions to create time and space for people to thrive.

#### 4.3.2. Threats to Professional Responsibility and Autonomy

Responsibility is a multi-faceted concept with implications for evaluating, sanctioning, and influencing people’s conduct and defined as “the quality or state of being responsible, such as moral, legal, or mental accountability” [36]. In turn, to be responsible carries elements of liability, causality, accountability, and answerability [37].

The extent to which individuals perceive responsibility depends on the psychological connection of three elements: identity images that apply to a person, standards of conduct that should guide a person’s behavior, and events that are relevant to the standards of conduct and the person [38]. Responsibility is judged high when there is a clear, well-defined set of standards for an event, a person is perceived to be bound by the standards by virtue of their identity, and a person has control over the event. In contrast, when standards are ambiguous or conflicting, of questionable pertinence to a person, or the person lacks control over an event, perceptions of responsibility are weakened.

Examples of standards among pharmacy personnel include pharmacy laws and regulations, professional codes of ethics, workplace policies, clinical guidelines, and practice norms among peers [39]. Such standards are internal role manifestations of an external social order, whereby the linkage between standards and a pharmacy personnel’s identity is their sense of professional duty or obligation to these roles that guide how they should act or are expected to act [40]. Control over an event encompasses varying levels that have relative contributions of internal forces, such as foreseeability and intent, and external forces, such as the environment in which an act takes place or in the case of omissions, does not take place [41].

That for which one is responsible and to whom one is responsible are largely dictated by the positions one holds in society. In this context, professional responsibilities of pharmacy personnel encompass a myriad of roles, both traditional and novel. Pharmacy personnel are responsible for and held accountable for roles ranging from dispensing medications to telehealth consultations, patient care decision-making, administering COVID-19 vaccinations, and performing COVID-19 tests [42]. Roles that pharmacy personnel fulfill or perceive they are expected to fulfill include the potential for role overload and role conflict. Role overload occurs when an individual performs multiple roles simultaneously yet lacks the resources to accomplish them, such as time, energy, and capabilities. Role conflict exists when two or more roles are contradictory, incompatible, or mutually exclusive [43]. Role overload and role conflict can lead to diminished perceptions of control and clarity of standards, both of which weaken perceptions of responsibility and autonomy.

Some of the study participants expressed threats to professional responsibility and autonomy that were characterized as disempowerment such as feeling deprived of power, authority, or influence. Several of these comments addressed lack of accountability among organizations that possess power to influence regulatory enforcement, educational accreditation standards, norms, and marketplace dynamics that shape the profession and pharmacy personnel’s working conditions:


*“The boards of pharmacy in every state should be out there visiting pharmacies and seeing what is really happening.”*



*“[The Accreditation Council for Pharmacy Education] needs to stop approving diploma mills and strip accreditation from the worst performing schools.”*



*“Regulatory capture of corporations infiltrating regulatory bodies, educational institutions, and advocacy organizations has effectively emasculated us as a profession.”*


The organizational actions, or inactions, expressed in the above comments magnify the lack of control that these pharmacy personnel have over these issues that are outside of their authority and influence.

The following comments by study participants were indicative of threats to professional responsibility and autonomy via power-dependence. This association implies that if pharmacy personnel commit or professionally invest in resources controlled by corporate pharmacies, they will be dependent on these pharmacies and have less power [44]:


*“Retail pharmacy has the potential to impact patients but are constantly being cut off at the knees with lack of pharmacist overlap and enough support staff making it almost impossible to reach that potential.”*



*“If [corporate pharmacies] truly cared about patients and safely serving them, they would invest in adequate staffing to ensure we do no harm…State boards and state/federal governments must reign in these atrocities for the sakes of patients and the health professionals held responsible for their care.”*


These comments touch upon the notion of the foreseeability to make errors or cause harm. When pharmacy personnel foresee potential harm, they feel responsible for preventing it. However, when they have little or no power to do so, their autonomy to prevent negative outcomes is threatened. Based on these findings, we propose that it would be fruitful to apply human factors and ergonomics systems approaches as outlined by Carayon and Perry [45] for deferring to local expertise, facilitating adaptive behaviors, enhancing system interactions along the patient care journey, re-purposing existing processes, and applying dynamic continuous learning. The goal for such redesigns would be to build resilience in the systems of care so that pharmacy professionals can quickly respond to tensions and disturbances to professional responsibilities.

#### 4.3.3. Learned Subjection

Learned subjection or subjugation results when an organization exerts both high levels of “control” over how work is done and “instrumentality” or the extent to which employees are treated as means toward an end [46]. This category was characterized by comments obtained from pharmacy personnel that represented the elements of oppression, abandonment, depersonalization and dehumanization. Pharmacy personnel expressed feelings of oppression or the inhibition of individuals’ ability to develop and exercise their capacities and express their needs, thoughts and feelings [47] in the following comments:


*“Between the lack of job options and the fear of losing our licenses, corporates/management/leadership figured they can basically increase workload to what it feels at this point indefinitely while they compensate for a set number of hours; I call it the “new slavery.”*



*“Feel like a slave, want to focus on patient care, but that’s the last thing we do if we get lucky.”*


Pharmacy personnel indicated that this type of oppression is characterized by workload that is ever increasing and uncompensated, the inability to focus on patient care, and market forces which lead to a perceived lack of job alternatives. Oppression entails perceptions of powerlessness, marginalization and exploitation [48]. Perceptions such as these may lead to what the nursing literature describes as “oppressed group behaviors” [49]. One such behavior is “silencing the self” which is manifested by not speaking about one’s own contributions to patient care or offering any feedback when asked [50]. These behaviors diminish nurses’ own sense of value and leads to feelings of marginalization, powerlessness and exploitation. Nurses in these situations are often fearful to express their needs which results in low self-esteem. Subjects in our study expressed many of these sentiments.

Feelings of abandonment also were exhibited in pharmacy personnel comments:


*“We are seriously struggling every day and no one seems to care.”*



*“Please do all you can to stop this. Pharmacists for retail pharmacies are literally dying on the job and [COMPANY] doesn’t care. They cannot keep getting away with this and I don’t want anyone else dying just because [LEADER] wants to make more money.”*



*“Yes, we all went into pharmacy to help people. But, when is someone going to help us?”*


Employees develop general perceptions concerning organizational support, which is the perceived extent to which an organization values their contributions and cares about their well-being [51]. Several meta-analyses confirmed that perceived organizational support is positively related to job satisfaction, affective organizational commitment and job performance [52,53]. Research on pharmacy personnel can be extended to investigate the concepts of abandonment and perceived organizational support.

Aspects of depersonalization also were expressed in pharmacy personnel comments:


*“As a pharmacist, I am treated as an easily replacement staff member with no respect for my education level or impact on the community.”*


Depersonalization has been studied in pharmacy as a component of burnout [4,54] and is defined as a dehumanizing response towards people who are the recipient of one’s services. While these studies show that pharmacists experience a high level of depersonalization, the measures used to describe depersonalization refer to pharmacists treating patients in a depersonalized manner and not to how pharmacy personnel feel when they are the recipient of this treatment.

An important aspect of depersonalization is dehumanization. Aspects of dehumanization were found in our analysis:


*“Not only do we not have time to take care of our patients with counseling and providing information with prescriptions as necessary, but we also have given everything of ourselves to try to meet all demands and our own health, mental and physical is waning.”*



*“They show no concern for my personal health at all.”*



*“Corporate does not look at our condition and say what can they do to help, but instead they criticize us for not meeting their goals and that we need to improve on all aspects of their ‘rubric’.”*


Organizational dehumanization is a perception which undermines one’s feeling of having a socially valuable existence [55,56] and arises amongst employees once they realize that they are being considered as a tool or a robot in their organization and can be replaced easily. Many of the comments describing both depersonalization and dehumanization are related to how they feel they are being treated by their employers. One respondent wrote:


*“I feel like I am treated like a robot, with high expectations of completion of all the tasks assigned to us, with many new tasks assigned constantly, but with no additional help given, yet they are constantly taking hours earned away.”*


Research is lacking in pharmacy personnel on these topics and their connection to well-being and resilience.

One final aspect found in the literature related to dehumanization and oppression is violence [57]. In our analysis we found interpersonal violence being perpetuated on pharmacy personnel by patients/customers:


*“Customers have nothing better to do than scream and yell at us and tell us we are incompetent and not fit to work there because of their scripts not being ready when they arrive. Something really needed to be done.”*


Bullying and harassment by customers/patients and co-workers was examined in the 2019 National Pharmacist Workforce Survey. In that analysis about 25% of pharmacists experienced some type of harassment [1]. When workers are frequently abused by their supervisors or others they could feel treated like less than human and could shift these negative perceptions to the organization [57]. Greater understanding of the negative effects of harassment and bullying in pharmacy workplace is needed.

Each of these elements has implications for the pharmacy workplace. Reactions such as these can cause pharmacy personnel to turn inward and suffer in silence. They are also not able to work together to take collective action to better well-being and resilience. This also inhibits the profession in moving forward. Additional work on these aspects is needed. We propose that more research into the aspects of learned subjection by pharmacy personnel including oppression, abandonment, depersonalization and dehumanization would help improve the pharmacy workplace. The findings suggest to us that pharmacy personnel have been mistreated and there is a need for restoration and renewal.

#### 4.3.4. Moral Distress and Moral Injury

Related to the previous section, the results further expose the experience of moral distress and moral injury in pharmacists and pharmacy personnel. One respondent stated,


*“The push to do so much more with less personnel, less trained and/or competent personnel has resulted in a need to come early, work late, work without eating, work without using the restroom…”*


The findings also contained expressions of fear of making ‘very serious mistakes at work’ or ‘worry about maintaining patient safety and providing the best healthcare’ given all of the clinical and technical activities required of pharmacists. Respondents wrote:


*“Giving everything of ourselves to try to meet all demands and our own health, mental and physical health is waning.”*



*“Practice today as “soul-killing”.*



*“I honestly contemplated suicide a few times this last year and no, that is not an exaggeration.”*


Pharmacist suicide must have our attention so that we can take action to deepen our understanding of whether moral injury (or post-traumatic stress disorder) contributes to this alarming outcome as identified in veterans [58,59,60]. It is noteworthy that Lee and colleagues [58] identified a higher suicide rate for pharmacists than the general population during the years of their analysis (2003 to 2018).

Moral distress and moral injury are areas of research for deepening our understanding of what pharmacy personnel experience in an array of patient and non-patient care settings. A conceptual framework has been proposed to illustrate the distinctions and overlap between moral injury and moral distress that may be useful to apply to pharmacy to answer a plethora of questions [61,62]. What factors contribute to the development of moral distress? What impact does moral distress have on the development of mental health issues or on the development of moral injury? Are there differences seen amongst those engaged in different direct patient care settings and non-patient care settings such as working in regulatory or medical affairs within a pharmaceutical company or making decisions pertaining to managing populations? Further, applying moral distress research to pharmacy can help inform what organizations and regulatory bodies need to do to enrich the work environment to enable pharmacy personnel to meet their responsibilities to their patients and communities.

Moral injury currently lacks consensus around a definition and literature in this area is in an early adolescence stage—especially in pharmacy. Identifying and adopting a conceptual framework that is applicable across different areas of the profession and integrating assessment as part of future pharmacy workforce studies would be useful for characterizing moral injury experienced by pharmacists. A recent critical literature review of moral injury proposes research needs that may have applicability to pharmacy such as determining the relationship between specific morally injurious events and outcomes; broadening examination of moral injury to pharmacy personnel; or considering how moral injury may amplify the risk for developing substance use disorders or exacerbating them [63]. It seems logical that moral injury contributes to suicidal ideation or that worry over committing medication errors leads to anxiety. However, it is necessary to discover whether specific associations exist. Pharmacy-specific research on moral injury is critical to learn how moral injury develops for pharmacy personnel. The determination of effective strategies for managing it and how to pressure-proof pharmacy personnel during their careers to mitigate its onset is essential.

#### 4.3.5. Sociocultural Effects

The findings affirmed that sociocultural effects, such as workism, are part of how pharmacy personnel describe their workplace wellbeing and resilience. Respondents wrote:


*“Please help, I can’t believe my career has come to this. I’ve become obsessed with work only to have it take over my life.”*



*“Help us. We are drowning. We are dancing as fast as we can, but somehow the platform gets wider and wider, our shoes smaller, and the music louder. No one can keep this up.”*


The sociocultural phenomenon of workism is an area of research for expanding our understanding of pharmacy workplace wellbeing and resilience. There is a need to develop innovative ideas for helping our society, including health care systems, evolve and mature in response to the need to help health care providers experience self-actualization outside of salaried jobs [22,23].

Another area for future inquiry that is related to workism is how large-scale reform can improve the experience of providing and receiving care. Halfon and colleagues [63] propose that there is need to integrate societal and health care system interests into the model of health care. That is, there is a need to advance from a coordinated health care system in which the focus on making the system more efficient, effective, and profitable to one that is community-integrated. Together, health care providers and patients can build community-integrated health systems that meet the needs of specific communities and use a bottom-up approach for developing systems of health optimization. This could be a way to overcome workism and the stress that comes from feeling trapped in one’s work.

Finally, we propose that including the quintuple aim for health care improvement is needed. The quadruple aim—improving population health, enhancing the care experience, reducing cost, and improving the experience of providing care—was enhanced in 2022 by Nundy and colleagues [64] with the pursuit of health equity being elevated as the fifth aim for health care. We propose that this sociocultural shift will be important for improving wellbeing and resilience in health care for both providers and for patients. Building the opportunity to attain one’s full health potential with no one disadvantaged from achieving this because of social position or other socially determined circumstances will change incentives from top-down, mechanized, assembly line processes back to incentives that reward intimate caregiving relationships.

We propose that such change is possible as patient-tailored virtual communication, genomic-customized treatments, collaborative engagement for patients as co-producers of value, and anticipation of needs already are being driven by technological advancements [65]. Research in the areas just presented will be fruitful and will help drive positive change for improving both patient and practitioner wellbeing.

#### 4.3.6. Plea for Health Systems Change

The previous sections have discussed the state of professional well-being for pharmacy personnel and described determinants of burnout and career satisfaction [66,67]. Evidence from respondents’ written comments suggests that individual-level interventions are inadequate for addressing the endemic issue of burnout in pharmacists [5,66,67], a gap that is perhaps best described by this quote from a 2018 article by Gregory and colleagues: “*Interventions to reduce burnout have sought to improve the resilience of an individual to withstand this [job demand–job resource] imbalance rather than identify and ameliorate the cause*” [68]. This realization informs the need for pharmacy professionals to embrace a systems orientation in their efforts to improve well-being in the pharmacy workspace. A systems framework conceptualizes pharmacy work environments as complex systems in which the resources, constraints, incentives, and demands interact. Organizations influence how employees engage with their work within the organization as well as the individual and organizational outcomes derived from work [69]. Three areas constitute the focus of best practices for a systems approach to improving pharmacy professional well-being [69,70]: (1) aligning professional and business roles, (2) work-system redesign and organizational learning, and (3) rethinking legislative and regulatory frameworks.

The pharmacy profession, especially in the community setting, has been struggling to settle the conflict between professional and business roles for over a century [71]. As mentioned previously, role conflicts are a source of dissonance that pharmacists experience at work. Corporate goals of efficiency and profit maximization and pharmacy professionals’ goals of patient care often do not align [72]. One respondent wrote:


*“12 years ago I was in a slow store and was greatly able to make an impact on my patients and their health. However, in more recent years I have been required to increase sales, profit, script count…”*


Studies have shown that when corporate and professional values align, health care professionals have a more positive view of company culture. This has been associated with improved work engagement, performance, and satisfaction among healthcare professionals [72,73]. Addressing role conflicts in pharmacy organizations requires all stakeholders to recognize the benefits of pharmacy professionals’ work in an effort to provide better care to patients and help them engage in work that aligns with their professional interests. Pharmacy organizations will benefit from finding common ground between process efficiency and profit maximization goals with pharmacy professionals’ need for joy and fulfillment at work [74]. This includes redesigning organizational structures, processes, and policies to prioritize meaningful work for personnel even if it may come at short term financial costs to the organization through the need for more resources and making job demands more in line with pharmacy personnel operating in a professional orientation. The long-term benefits are expected to include employee retention and better quality care, which will be important for pharmacy to continue to move toward being reimbursed for providing value to health systems.

As alluded to in the previous paragraph, redesigning organizational work processes and systems begins with improving the nature of the work that pharmacy professionals do. Excessive workload, monotonous and low-value work that do not require personnel to use their skills and training to the fullest extent are major contributors to burnout [5,75]. One respondent wrote:


*“Pharmacists are doing the best they can but the workload we are juggling is an error waiting to happen.”*


Physical aspects of pharmacy workplaces need to be redesigned by incorporating principles from human factors, ergonomics, and human-centered design to reduce occupational stress and fatigue [76,77]. Qualitatively, balance must be maintained between job demands like workload, length of work shifts, and job resources like organizational support, teamwork, and appropriate staffing [5,78]. Organizations have a duty to involve pharmacy professionals in decision making, development of performance metrics, and performance evaluations in a feedback loop that fosters organizational learning. Research shows that such a collaborative approach reduces the chasm between management and staff, as the staff are more likely to view leadership as authentic [79].

While pharmacy professionals and organizations may not be lacking in intent to implement professional well-being systems at work, it appears that such efforts often fall short of their objectives without effective legislative and regulatory frameworks. It was the view of many respondents in the present study that role conflict and work stress persist due to a lack of external mechanisms to hold pharmacy organizations accountable. One respondent wrote:


*“Pharmacy chains are promoting dangerous workplaces, both for the patients and employees. I sincerely hope state legislators or federal regulators step in before there is some horrible tragedy.”*


There is already enough tragedy in the work life of pharmacy professionals to warrant urgent legislative and regulatory interventions. Areas of pharmacy work life that can be targeted for regulatory intervention include ensuring effort-reward balance to deter performance metrics and compensation schemes that encourage quantity over quality of work done [80]. To a greater extent, there is a need for effective alignment of incentives to drive the desired system changes. Pharmacy professionals have noted that while the managerial pharmacist-in-charge (PIC) may be the representative of a pharmacy organization, as one respondent put it:


*“The PIC is responsible for everything but has no control over anything such as training, staffing, hours, budget… yet the boards go after the PIC.”*


Additionally, pharmacy professionals and organizations need better representation in legislation making. The future pharmacy workforce should also be educated about the challenges in pharmacy workplaces with a view to identifying opportunities for improvement.

## 5. Conclusions

There is shared malaise surrounding a collective experience of helplessness, loss of professional standing, oppression, moral injury, workism, and a plea for rescue. The results show the need for health care system and societal change and provide insights for future research.

We propose that there is a need for more expansive and extensive research that is specific to pharmacy workplace wellbeing and resilience. One area we uncovered in this study relates to the construct of “shared trauma.” Tosone and colleagues [35] outlined terms such as burnout, compassion fatigue, secondary traumatic stress, vicarious trauma, and counter-transference. These resonate with our findings and merit further study for how systems can be changed to help pharmacy professionals have or maintain the capacity to care for patients in a compassionate manner.

Another domain for inquiry relates to “professional responsibility and autonomy” for assuring patient care and medication safety. We propose that human factors and ergonomics systems approaches [45] would serve as useful foundations upon which to base further research. Carayon and Perry [45] outlined the importance of systems that are deferential to local expertise, adaptable, interactive, based on existing processes, and dynamic with continuous learning. Such systems would improve professional responsibility and autonomy for patient care and medication safety.

“Learned subjection” was another theme that emerged in the findings. Several research domains were identified that could be useful for future work including: social justice [46,47,48], challenges observed in nursing [49,50], organizational support [51,52,53], and psychology [55,56,57]. In a related manner, “moral distress and moral injury” research is rooted in the psychology literature as well [58,59,60,61,62]. Learning more about possible links among learned subjection, moral distress, moral injury, anxiety, suicide, medication errors, and substance use disorders is necessary. Relatively little is known about these in the pharmacy domain. Based on the findings from this study, we propose that these topics deserve more attention.

In addition to research that is focused on individuals, more work is merited for “sociocultural effects.” Our findings revealed that the sociocultural phenomenon of workism [22,23] is negatively affecting pharmacy workplace wellbeing and resilience. Activities focused on building community-integrated health systems that meet the needs of specific communities and applying a bottom-up approach for developing systems of health optimization could be helpful. Halfon and colleagues [63] proposed that there is a need to integrate societal and health system interests into a new model of health care. We propose that such a sociocultural shift would improve the wellbeing and resilience for both providers and for patients.

The findings showed the need for “health systems change,” another form of systemic support. Current systems are inhumane in that they are creating burnout [68]. Hence, we propose the application of a systems framework [69,70] in pharmacy. Some work has been reported in the physician and nurse domains [68,69,77] and application in pharmacy is justified. Specifically, there is a need for (1) aligning professional and business roles, (2) work-system redesign and organizational learning, and (3) rethinking legislative and regulatory frameworks [69,70].

In summary, business models driven by mechanized assembly line processes, business metrics that supersede patient outcomes, and reduced pharmacy professional judgement have contributed to the decline in the experience of providing patient care in today’s health systems. The results showed that the portrait of respondents’ lived experiences regarding pharmacy workplace wellbeing and resilience is beyond the individual level. We propose several areas for expanded inquiry in this domain: (1) shared trauma, (2) professional responsibility and autonomy, (3) learned subjection, (4) moral injury and moral distress, (5) sociocultural effects, and (6) health systems change.

## Data Availability

Data used for this study is stored in electronic format and may be obtained from the corresponding author at schom010@umn.edu.

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
