# Peer review of "Pharmacy Workplace Wellbeing and Resilience: Themes Identified from a Hermeneutic Phenomenological Analysis with Future Recommendations"

_pharmacy, 2022, doi:10.3390/pharmacy10060158_

Round 1

Reviewer 1 Report

This is an extremely timely article as stress and burnout are the number one issues on pharmacists' minds right now.  Analysis of pharmacists' comments from the survey and providing context surrounding these comments will be useful in initiating change and encouraging further research surrounding these issues.

Author Response

We greatly appreciate the reviewer's comments and support for this work. 

Reviewer 2 Report

Thank you for the opportunity to review this important paper. I applaud the authors for making meaning of and drawing themes from the 2021 APhA/NASPA National State-Based Pharmacy Workplace Survey. The findings from this hermeneutic phenomenological method are vital for the profession and our patients. To be honest, this was one of the most depressing papers I've read in 2022 and also one of the most important.

This paper is a significant contribution to the literature, well organized, scientifically sound and appropriately referenced. The narrative created by the authors' analysis of comments bring to life the real story of pharmacy workplace despair that statistics from the national survey are unable to convey. I offer the following comments for your consideration.

1. The title, while succinct, doesn't capture the essence of this paper. Perhaps a subtitle such as Pharmacy Workplace Wellbeing and Resilience: Shared trauma, moral distress and additional themes identified from a phenomenological approach to written comments from the 2021 APhA/NASPA National State-Based Pharmacy Workplace Survey.

2. How are the 614 written comments representative of the various practice settings identified from the respondents? In 4.0 Discussion/4.1 Limitations, line 276 states that "The majority of comments came from pharmacy personnel working in community practice." Please elaborate. Perhaps, a supplemental table with all 17 practice settings identified in the original survey and a count of all 6973 responses stratified by practice settings in one column and then a count of the 614 comments by practice setting in another column. This would help to show which practice settings were over/under-represented among the comments compared to the original survey responses.

3. In section 2.0 Materials & Methods/2.1 Data Source, line 152 and 153, "...usable responses were from those who wrote an in-depth comment detailing stories and experiences related to pharmacy workplace wellbeing and resilience." Please elaborate on what "in-depth" means and how the 614 comments were selected from the 6,973 responses. Were criteria used to determine when a comment was "in-depth"? 

4. In Section 3.0 Results, 15 elements and 6 categories were identified. Were all themes negative? Were there any positive themes related to wellbeing and resilience?

5. Section 4 Discussion/ 4.3.4 Moral Distress and Moral Injury, lines 532-534 regarding pharmacist suicide, the article (59) by Lee et al is cited. It may also be important to add that their longitudinal analysis of suicides among pharmacists identified a higher suicide rate than the general population.

Lastly, thank you for revealing and sharing these important findings in your paper!

Author Response

We greatly appreciate the reviewer's comments and suggestions for improving the paper. We addressed the suggestions as follows:

  1. We updated the title using a subtitle as suggested. It is more informative now. If editors or reviewers have suggestions for changing the title even more, we are certainly open to that.
  2. Based on the suggestions about practice settings being over/under-represented, we added more information to the Limitations section. Interested readers have been instructed to refer to the full report for the study (Reference 9). That report provides in-depth information about the distribution of respondents and the associations between variables and respondent types.
  3. Section 2 has been updated as suggested. The methods now include a description of the inclusion criteria we used for selected the 614 comments. We base the criteria on references 24-29 and included those in the section.
  4. Section 3, Results. As suggested, we now describe how positive comments were few and they related to how these respondents were thankful where they work so that they can avoid one of the 15 elements we identified that they hear about from their colleagues working in other places.
  5. Section 4, Discussion. As suggested we added information about how pharmacists have been identified as having higher suicide rates than the general population (based on Lee, et al.'s work; reference 59).
  6. Taken together, all of the comments have provided better flow and better clarity for the paper. Thanks again for these great suggestions.